# DeRL: Diverse-Exploration Reinforcement Learning for Large Language Models Improves Mathematical Reasoning

## Abstract

Current reinforcement-learning (RL) pipelines for large language models (LLMs) that tackle mathematical reasoning and formal theorem proving tend to over-exploit a few high-probability chain-of-thought (CoT) sequences. Because rewards are granted solely for producing correct answers, the policy quickly converges on those paths, neglecting the rich space of alternative proofs and solution strategies that math problems usually have. We address this limitation with Diverse-Exploration RL (DeRL), a simple yet effective modification to the reward function and the RL prompts. During training, the model is explicitly instructed to solve each problem without relying on its previously generated CoT. Next, an auxiliary LLM judge verifies the approach dissimilarity between the new LLM output and the previous CoT. Combined with the correctness metric, this new reward encourages exploration of novel reasoning paths while preserving accuracy. We test DeRL on both natural-language math questions with boxed answers and formal theorem proving problems in Lean. Across the MATH benchmark and Leanabell dataset, DeRL yields more than 10% relative gain compared to the PPO baseline for the Pass@1 metric. DeRL also consistently yields better results for the Pass@N metric. Our findings demonstrate that incorporating diversity-aware rewards facilitates broader exploration and enhances reasoning capabilities of LLMs, indicating a promising direction for improving current reinforcement learning pipelines.

## 1 Introduction

Large language models (LLMs) have demonstrated strong capabilities in solving complex reasoning problems, achieving impressive performance on natural language questions with boxed answers (DeepSeek-AI et al. (2025); Yang et al. (2025); Team et al. (2025)). Beyond natural language reasoning, LLMs have also shown promise as theorem provers and tactic generators for solving formal theorem proving tasks in environments such as Lean. (Moura & Ullrich (2021); An et al. (2024); Ren et al. (2025) ).

A key factor behind this success is the use of reinforcement learning, which has proven particularly effective in fine-tuning LLMs to enhance their reasoning abilities.(DeepSeek-AI et al. (2025)). Current RL pipelines for model post-training on reasoning tasks consist of the following steps: prompt the model with the problem statement, have the LLM generate CoT steps, and then verify the final answer against the ground truth (for natural language problems seeking a boxed answer) or use a proof assistant (such as Lean) to check the correctness of a formal proof (Wang et al. (2025); DeepSeek-AI et al. (2025); Grattafiori et al. (2024)). The reward is backpropagated to the entire CoT solution, reinforcing model behavior on producing the CoT that led to an acceptable answer during RL training.

A key challenge arises during the reinforcement learning (RL) phase: once the model encounters a problem and receives a reward for producing a particular chain-of-thought (CoT) solution leading to a correct answer, it tends to over-commit to that path. When presented with the same problem again, the model is less likely to explore alternative reasoning strategies. This behavior contradicts the nature of mathematical problems and theorems, which often admit multiple semantically distinct solution methods. For instance, a number theory problem may be approached from algebraic, analytic, or geometric perspectives—each demanding fundamentally different intuitions. These

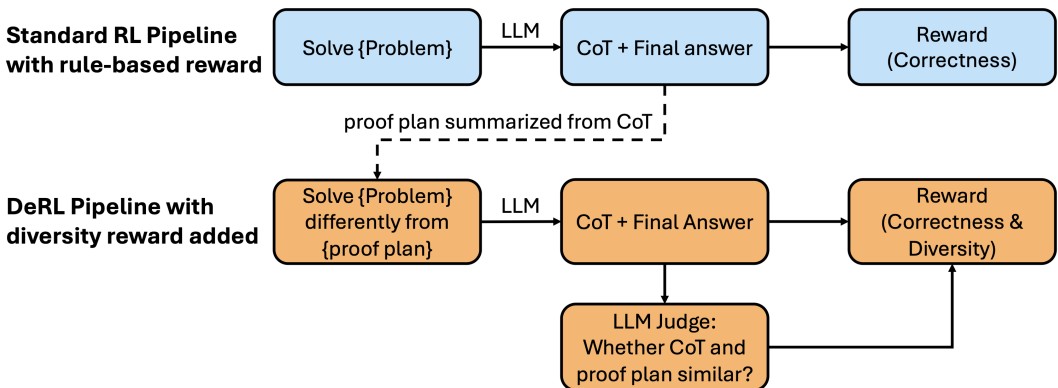

Figure 1: Overview of the DeRL Pipeline vs. Standard RL with Rule-Based Reward. In DeRL, the model is explicitly instructed to avoid generating CoTs similar to its own prior rollouts from the standard RL phase. The reward signal combines both correctness and diversity to encourage exploration of alternative reasoning paths.

diverse approaches frequently reflect entirely different mathematical insights, the discovery of which requires distinct modes of understanding.

Due to the lack of incentive for diverse exploration, several studies have raised concerns about the limitations of current reinforcement learning pipelines. For instance, Yue et al. (2025) argues that these approaches primarily improve Pass@1 performance, while offering limited gains for Pass@$n$ when $n$ is large. In the context of formal theorem proving in Lean, Wang et al. (2025) observes that LLM-generated proofs often rely heavily on automation tactics such as `omega` and `linarith`, indicating a lack of diversity in tactic generation and suggesting a superficial level of reasoning.

In order to address this low diversity problem of the current RL pipeline, we propose to incorporate a diversity-encouraging reward into the RL process. In our experiments on natural language problems with boxed answers, we conduct two RL phases per problem. In the first, the pre-trained model is fine-tuned using only correctness-based rewards. In the second, the model is prompted with both the problem and a summary of its previous CoT, and instructed to avoid that method. An LLM judge (Claude-3.7) evaluates semantic similarity: if the new CoT resembles the previous one, the model receives zero reward; if it is both dissimilar and correct, it receives a reward of one.

For theorem proving in Lean, we observe that LLMs frequently rely on a set of automated tactics – such as `linarith`, `simp_all`, `ring`, `aesop`, `nlinarith`, `omega`, `simp`, `trivial`, and `positivity`—which appear with high frequency in the pretraining data. In the Leanabell dataset used for both RL and evaluation (Zhang et al. (2025)), over 80 percentage of proofs contain at least one of these tactics. These tactics invoke automated search procedures, helping the model to bypass the construction of explicit proofs that require genuine mathematical reasoning. We assume that rewarding the model to produce Lean proofs that do not contain any tactics from this list will induce stronger theorem proving ability overall. The model will be forced to address portions of the proof that it would otherwise be able to skip, broadening its output distribution to include additional techniques.

Experimenting with 7B models, we find that DeRL improves model performance over the PPO baseline on both natural language problem solving with boxed answers and formal theorem proving in Lean. Additionally, the diversity metric effectively encourages the generation of varied solutions. For the natural language math problems, over the course of DeRL training models achieve increasingly higher solution diversity scores on the training set. For Lean theorem proving, the proportion of automation-tactic-free proofs in the test set consistently increases throughout training. This improvement in solution diversity, even when independently considered from potential gains in accuracy, is valuable as it promotes training the model to be able to find multiple distinct methods for solving a given problem. This is particularly desirable for exploratory downstream applications where we want a model to try diverse approaches across repeated inference passes, like in automated theorem proving.

Our major contributions are,

- We propose our RL pipeline, DeRL, with a simple and model-agnostic diversity reward, which encourages a model to explore diverse paths in the solution space. We note that our method remains orthogonal to the choice of RL algorithm used in practice.
- We demonstrate consistent gains of DeRL on two natural and formal language reasoning tasks with 7B models, compared to a standard PPO baseline measured by Pass@1 and Pass@N.
- We show that DeRL promotes the diversity of LLM rollouts during training and testing.

## 2 RELATED WORK

**Theorem proving in formal languages**.     The intersection of large language models (LLMs) and interactive theorem provers has rapidly evolved from proof-of-concept demonstrations to systems that solve thousands of benchmarks autonomously. GPT-f first showed that a transformer can emit Lean tactics that close simple goals, establishing a template for LLM–assistant interaction (Polu & Sutskever (2020)). Subsequent work introduced co-training schemes that alternate between synthesizing proofs and retraining the model on its own successes, leading to steadily deeper search trees and higher proof rates (Han et al. (2022)). Lean has emerged as a focal environment thanks to its rich mathlib library and a compiler that provides fine-grained feedback on failed proof attempts (Moura & Ullrich (2021)). Modern systems treat the LLM as a tactic generator whose suggestions are verified (or rejected) by the Lean kernel, closing the supervision loop at every step (Yang et al. (2023)). Retrieval-augmented approaches further boost success by letting the model cite previously proven lemmas from mathlib or an accumulated proof memory (Lin et al. (2025a)). Correct search attempt, together with failed search attempt and the backtracking steps are found to be helpful when training LLM-based tactic generator(An et al. (2024)). Natural language reasoning has been incorporated into tactic generation to improve model performance on Lean theorem-proving tasks (Wang et al. (2025)). Together, these lines of work push the frontier from toy theorems toward curriculum-scale formal mathematics.

**Reinforcement learning for LLM reasoning**.     Reinforcement learning (RL) has increasingly established itself as one of the most important post-training paradigms for enhancing the reasoning capabilities of large foundation models. Unlike supervised fine-tuning, which primarily aligns model outputs with human-curated data, RL introduces a flexible optimization framework that allows models to adapt to complex objectives specified through reward functions. A prominent example is DeepSeek-R1, which demonstrated that a carefully designed reward signal—combining answer correctness with length penalties and tool-use bonuses—when paired with a standard RL pipeline, can yield substantial improvements on natural-language mathematical reasoning benchmarks (DeepSeek-AI et al. (2025)). This approach has proven influential. Notably, several flagship models, such as Qwen 3 (Yang et al. (2025)), Llama-3-Herd (Grattafiori et al. (2024)), and Gemma 3 (Team et al. (2025)), have incorporated RL-based post-training strategies to strengthen their performance on reasoning-intensive tasks. Collectively, these developments underscore RL's central role in shaping the next generation of foundation models with advanced problem-solving capabilities, and other generation tasks.

In the domain of formal theorem proving, reinforcement learning (RL) has emerged as one of the dominant strategies for training Lean proof generators. Recent systems have demonstrated the effectiveness of RL in this setting. For instance, DeepSeek-Prover-v2 employs an RL algorithm that directly rewards successful proof generation, enabling it to achieve state-of-the-art proof rates on the challenging MiniF2F benchmark (Ren et al. (2025); Zheng et al. (2022)). Similarly, Kimina-Prover integrates RL with reasoning-oriented chain-of-thought methods to improve its performance on Lean theorem-proving tasks (Wang et al. (2025)). These results highlight RL as a key paradigm for advancing automated reasoning in proof assistants.

Despite these successes, recent analyses highlight a tendency for RL-finetuned models to over-exploit the high-probability trajectories inherited from their supervised-finetuning (SFT) initialization. Havrilla et al. (2024) argue that such models rarely depart from memorized chain-of-thoughts, limiting true reasoning diversity. Yue et al. (2025) further question whether standard reward formulations genuinely *incentivize* new reasoning skills or merely reinforce surface-level patterns already present in the base model. Addressing this exploration–exploitation imbalance—through diversity-aware

rewards, curriculum design, or uncertainty-guided sampling—remains an open research challenge and a key motivation for the present work.

# 3 METHODOLOGY

## 3.1 RL PROMPTS PREPARATION

For the natural language reasoning tasks, for each problem statement in the 60k OpenMathInstruct datasets, there are two types of RL prompts as shown in the following box

> **RL Prompt Formats**
>
> The original RL prompts are the ones with the format
>
> > ```
> > {Problem statement} + solve the problem step
> > by step and put the final answer in the box as
> > \box{final answer}.
> > ```
>
> The diversifying RL prompts are the ones with the format
>
> > ```
> > {Problem statement} + {Proof plan} + I want
> > you to solve the problem in a way that is very
> > dissimilar to this proof plan.  If you solve the
> > problem in a way similar to the proof plan, that
> > is cheating.  Solve the problem step by step and
> > put the final answer in the box as \box{final
> > answer}.
> > ```

The generation process of the proof plan as used in the diversifying RL prompts will be explained in the Section 5.1.

For the formal language reasoning task, for each problem statement in the 60k Leanabell training dataset, there are two versions of RL prompts as shown in the following box.

> **Lean RL Prompt Formats**
>
> - The original RL prompts are the ones with the format:
>
>   > ```
>   > Complete the lean code + {Lean problem
>   > statement}
>   > ```
>
> - The diversifying RL prompts are the ones with the format:
>
>   > ```
>   > Complete the lean code + {Lean problem
>   > statement} + Note you absolutely cannot use any
>   > tactic from the list ['linarith', 'simp_all',
>   > 'simp', 'omega', 'ring', 'aesop', 'positivity',
>   > 'trivial'], otherwise it is cheating and the
>   > proof you found is invalid.  + {Lean problem
>   > statement}
>   > ```

## 3.2 METRICS

### 3.2.1 CORRECTNESS METRIC

For both the natural and formal language reasoning tasks, there are two metrics. The first metric is the classic correctness metric. The model prompted with the natural language math problem is expected to generate a CoT solution with a boxed final answer. If the generated final answer in the box string matches with the ground truth, then the correctness metric in the context of natural language reasoning will assign 1 as the reward. Otherwise it assigns 0. As for the formal task, the model prompted with the math problem statement in Lean and is expected to generate a series of tactics

which serve as the proof of the statement. The statement and the generated tactics are both sent to the Lean server, which will then return True or False based on whether the given tactics successfully prove the theorem statement. The correctness metric in the context of formal language reasoning will assign 1 if the Lean server returns True and 0 otherwise.

### 3.2.2 DIVERSITY METRIC

The second metric is the diversity metric. For the natural language reasoning task, there is a proof plan given in the diversifying RL prompt. After the model is prompted with the diversifying RL prompt and generates the CoT path with the final answer in the box, an external LLM (Claude-3.7 in this paper) is used to check if the generated CoT path is semantically similar to the proof plan in the prompt. The external LLM is prompted with

---

**LLM judge prompt**

You are a careful math teacher. I will give you the problem statement, the solution of the student, and a proof plan.
The student has been taught with the proof plan, and he is instructed to find a method to solve the problem that is very different from the proof plan.
I want you to carefully check if the solution of the student is very similar to the proof plan.

**Problem statement:** {prompt}
**Proof plan:** {proof_plan}
**Student solution:** {solution_str}

Output ***True*** if the solution is similar to the proof plan (so the student cheated). In all other cases, output ***False***.

---

For the natural language reasoning task, the diversity metric assigns a reward of 1 if both of the following conditions are met: (i) the external LLM judge outputs False (indicating semantic dissimilarity to the reference CoT, and (ii) the final boxed answer produced by the model exactly matches the ground truth. If either the judge outputs True (indicating similarity), or the final answer is incorrect, the reward assigned is 0.

For the formal language reasoning task, the model is given a diversifying RL prompt with the explicit instruction: "Note you absolutely cannot use any tactic from the list ['linarith', 'simp_all', 'simp', 'omega', 'ring', 'aesop', 'positivity', 'trivial']; otherwise, it is cheating and the proof you found is invalid." The model is expected to generate a proof using Lean tactics. If any tactic from the forbidden list appears in the generated proof, the diversity metric assigns a reward of 0. If no tactic from the list appears and the generated proof is verified successfully by the Lean server (i.e., it proves the given statement), the diversity metric assigns a reward of 1. We do not reward proofs that fail to be verified, even if they avoid all tactics in the automation list.

Note that the diversity metric will only be applied to the diversifying RL prompts to generate rewards.

### 3.3 RANDOM REWARD

As a baseline for comparison against the correctness and diversity reward metrics, we include a random reward, in which a reward of either 0 or 1 is assigned uniformly at random to the model outputs during RL training.

### 3.4 WHY WE USE PPO INSTEAD OF GRPO?

DeRL is orthogonal to the choice of RL algorithm used in practice. In our experiments, we choose PPO over GRPO. GRPO has been widely adopted in LLM post-training for reasoning tasks. However, it requires generating multiple rollouts per prompt—typically 8 or 16—because its reward function relies on assessing the relative quality among these rollouts. Our DeRL framework leverages an external LLM judge (Claude-3.7-Sonnet in our case) to evaluate the dissimilarity between a newly generated chain-of-thought (CoT) and a reference CoT. When the number of rollouts n is large, this

| Method | Qwen-MATH | DeepSeek-MATH | Goedel-Leanabell |
|---|---|---|---|
| Baseline | 0.507 | 0.402 | 0.710 |
| Standard PPO | 0.614 | 0.417 | 0.767 |
| Random Reward (on all prompts) | 0.582 | / | / |
| Partial Random Reward | / | 0.414 | / |
| DeRL Non-LLM-Rollout 4:1 | 0.615 | 0.422 | / |
| DeRL 2:1 | 0.615 | 0.420 | 0.768 |
| DeRL 4:1 | **0.623** | **0.430** | **0.778** |

Table 1: Pass@1 performance on natural and formal language reasoning tasks. Results are shown for models trained using PPO-baseline, DeRL, and random reward. For each method, the best-performing checkpoint across the entire training epoch is reported. In DeRL experiments, different mixtures of original and diversifying RL prompts are used. For Partial Random Reward row, random reward is only applied to the diversifying prompts. The correctness metric is still applied to the original RL prompts in this case.

leads to substantial API costs due to repeated external evaluations. To mitigate this, we adopt PPO instead of GRPO, setting $n = 1$, which significantly reduces the number of required LLM judge calls.

## 4 EXPERIMENTAL SETTINGS

### 4.1 DATASET AND MODELS

To evaluate DeRL on natural language reasoning tasks, we use the test split of MATH (Hendrycks et al. (2021)) as the test set. For training, we use the problem statements and the boxed answers from `nvidia/OpenMathInstruct-2` as RL prompts and the ground truth against which we verify LLM output (Toshniwal et al. (2024)). Note that this dataset also contains the CoT path leading toward the final solution. `nvidia/OpenMathInstruct-2` is a synthetically generated natural language dataset with difficulty and format similar as those of MATH dataset. It's important to note that `nvidia/OpenMathInstruct-2` has been decontaminated from the test split of MATH. We use the first 60k problems as the RL prompts.

We use `stoney0062/Leanabell-Prover-Traindata-SFT` (referred to as Leanabell) as the training and test set to evaluate DeRL on formal reasoning tasks (Zhang et al. (2025)). Leanabell contains 219 000 high-school–level mathematics problems statements and proofs in Lean. We take the problem statements from the first 60 000 problems as RL prompts and reserve the final 1000 for testing.

For natural language reasoning experiments, we use `Qwen/Qwen2.5-7B` and `deepseek-ai/deepseek-math-7b-instruct` as the base models (Yang et al. (2025); DeepSeek-AI et al. (2025)). For formal language reasoning experiments, we use `Goedel-LM/Goedel-Prover-SFT` as the base model (Lin et al. (2025b)).

### 4.2 REINFORCEMENT LEARNING AND HYPERPARAMETERS

We use VeRL (Sheng et al. (2024)) to perform reinforcement learning training and evaluation inference, employing the standard Proximal Policy Optimization (PPO) algorithm with $n = 1$ (Schulman et al. (2017)) and VLLM (Kwon et al. (2023)); The model generates a single rollout per prompt. We train the model using two compute nodes, each equipped with 8 NVIDIA A100 GPUs, for a total of 16 GPUs with 40 GB of memory per GPU. The training uses a global batch size of 512, with each mini-batch consisting of 64 examples. Each individual GPU processes a batch of 4 examples per optimization step. In all of the experiments, the models are fine-tuned for one epoch.

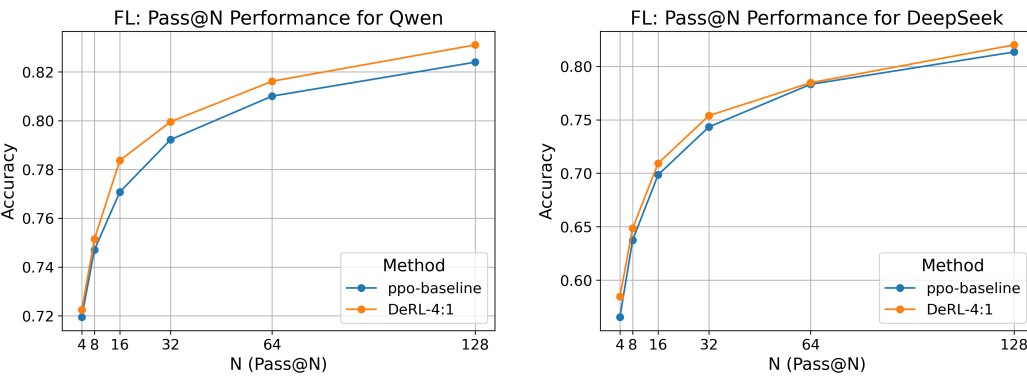

(a) Pass@N performance for Qwen on MATH      (b) Pass@N performance for DeepSeek on MATH

Figure 2: Pass@N performance for the natural language reasoning task. For both ppo-baseline and DeRL 4:1 experiments, We use the checkpoint that has the highest Pass@1 performance during the 1-epoch training of the model. Temperature for all Pass@N experiments is set to be 0.7.

## 5 MAIN RESULTS

### 5.1 EVALUATING THE IMPACT OF DERL ON MODEL REASONING PERFORMANCE

To evaluate the effect of DeRL on the reasoning capabilities of the LLMs, we perform DeRL on the Qwen, DeepSeek and Goedel models for natural and formal language reasoning tasks. For the natural language reasoning tasks, we begin by fine-tuning the Qwen and DeepSeek models on the 60k data samples with the original RL prompts for 3 epochs using standard PPO with rewards provided by the correctness metric. We then collect all chain-of-thought (CoT) reasoning paths that lead to correct final answers. If there are multiple CoTs leading to correct final answers for a single RL prompt, we only keep the first CoT. Using these successful trajectories, we prompt DeepSeek-R1 to generate a summary, referred to as the proof plan, following the format illustrated in the following Box 5.1.

> **Prompt for Generate proof plan**
>
> I will give you a math problem and a chain-of-thought solution of the problem. Please do the following thing:
> Summarize the solution of the problem in short sentences. Remember, no number should be shown in the summary. Don't give out the final solution.
> The goal for summarizing is to convey the general idea of the proof, and hide the calculation detail.
> Put the summary enclosed by `$$$`, as `$$$summary$$$`.
> Problem:
> `Prompt`
> Chain-of-thought solution:
> `CoT path`

Next, the proof plan is incorporated into the diversifying RL prompts, as outlined in Section 3.1. This yields a total of 60,000 original RL prompts and over 30,000 diversifying RL prompts. To construct the training set for DeRL in the natural language reasoning setting, we combine the original and diversifying RL prompts in ratios of 2:1 and 4:1, resulting in training sets containing 90,000 and 75,000 prompts, respectively. Standard PPO is then used to train the Qwen and DeepSeek models on these mixed datasets. As described in Section 3.2, the correctness metric is applied to the original RL prompts to generate the reward, while the diversity metric is applied to the diversifying RL prompts to generate the reward.

For the formal lanuage reasoning task, the situation is simpler. We skip the proof plan preparation stage since there is no proof plan in the diversifying RL prompts for the theorem-proving task, as shown in Section 3.1. To construct the training set for DeRL in the formal language reasoning setting,

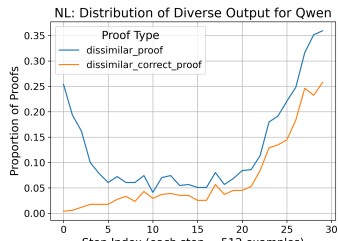 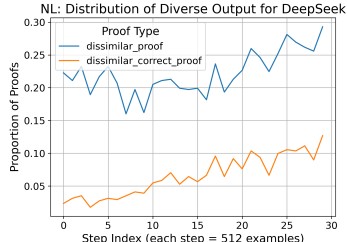 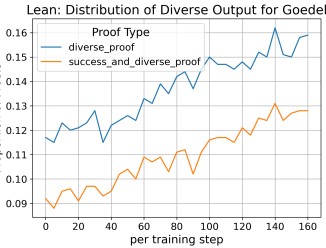

(a) Measured on training set: Diversity of the LLM output for Qwen DeRL 4:1

(b) Measured on training set: Diversity change of the LLM output for DS DeRL 4:1

(c) Measured on test set: Diversity change of the LLM output for Goedel Lean DeRL 4:1

Figure 3: Diversity change of the LLM output trained with DeRL during training. For (a), the point (25,0.149) on the orange line means that after model has been progressively trained on $25 \cdot 512$ many diversifying RL prompts during DeRL 4:1 training (model has also been trained on about $4 \cdot 25 \cdot 512$ original RL prompts at this point), out of the 512 LLM rollouts generated for the last 512 diversifying RL prompts, there are 14.9 percent of those LLM rollouts that both have correct final answers and are diverse as judged by the external LLM judge. For (c), the point (120, 0.121) on the orange line represents that after model has been trained with DeRL 4:1 for 120 steps, out of the 1000 proofs it generated for the 1000 test cases it is evaluated on, 12.1 percent of the proofs are both diverse and correct.

we combine the original and diversifying RL prompts in ratios of 2:1 and 4:1, resulting in training sets containing 90,000 and 75,000 prompts, respectively. Standard PPO is then used to train the Goedel model on these mixed datasets. Again, as described in Section 3.2, the correctness metric is applied to the original RL prompts to generate the reward, while the diversity metric is applied to the diversifying RL prompts to generate the reward.

**Models trained with DeRL perform better on the test set compared to standard PPO baseline**. As shown in Table 1 and Figure 2, for both natural and formal reasoning tasks, models fine-tuned with DeRL using a 4:1 ratio of original to diversifying RL prompts consistently outperform the PPO baseline with Pass@1 and Pass@N metrics, where the PPO baseline relies solely on the correctness metric as its reward signal. The temperature for all Pass@N experiments is set to 0.7. Notably, the maximum number of training steps across all three graphs is 150, corresponding to one full epoch of training under the 4:1 data mixture (75,000 data samples in total, with a training batch size of 512). For the natural language reasoning tasks, we additionally include random reward experiments for both the Qwen and DeepSeek models under varying random reward settings. In these experiments, both models are trained with the same 4:1 prompt mixture. For Qwen, we assign random binary (0/1) rewards to both the original RL and the diversifying prompts, replacing both the correctness and diversity metrics. For DeepSeek, we retain the correctness-based reward for original prompts but assign random binary rewards to the diversifying prompts in place of the diversity metric. As shown in Table 1, introducing random rewards leads to improved performance for the Qwen model, consistent with findings from Shao et al. (2025). However, the model trained with properly defined correctness and diversity-based rewards still significantly outperforms its random-reward counterpart. On the other hand, results in Table 1 show that applying random rewards to the diversifying RL prompts (correctness metric still applied to the original RL prompts) degrades performance for DeepSeek, with the PPO baseline outperforming the random-reward variant in this case.

**Diversity of the output of the model increase progressively over the course of training**. To evaluate the impact of DeRL on model output diversity, we track the diversity metric reward during training on the natural language reasoning task using the 4:1 DeRL run. Specifically, for every batch of 512 diversifying RL prompts during training, we compute the proportion of generated chain-of-thoughts (CoTs) that the external LLM judge deems semantically dissimilar from the proof plan in the prompt. Among these, we further measure the proportion that also produce a correct final boxed answer. The results are presented in Figure 3a and 3b. For the formal reasoning task, we evaluate the model checkpoint on the 1k-sample Leanabell test set every 5 training steps during the DeRL 4:1 run. For each evaluation, we compute the proportion of proofs that do not include any tactic from the automation tactic list defined in the introduction (see List 1), and proofs that are both

checked by the lean server and free of the automation tactics. As shown in Figure 3, the diversity of the LLM output is increasing in both NL and FL experiments, showing that DeRL successfully encourages the model to explore more diverse output during training and testing.

**DeRL Works Best When Diversifying Prompts Use Proof Plans from the Current Learner**. We assess the robustness of DeRL when the proof plans used to build diversifying prompts are obtained from a model other than the learner itself. In our natural language reasoning experiments, the proof plans were extracted from the CoTs that originally come from the dataset `nvidia/OpenMathInstruct-2`, which are produced by Llama-3-405B. The models that are fine-tuned in DeRL are Qwen and DeepSeek models. The results are shown in the row named DeRL Non-LLM-Rollout 4:1 in Table 1. As shown in the table, this mismatch diminishes the gains from DeRL: the learner is asked to deviate from a largely arbitrary proof trajectory instead of from its own high-probability strategy (that is generated in the previous RL round), hence weakening the exploratory signal on which DeRL uses to drive the learner toward genuinely novel reasoning paths.

**Too much diversifying prompt hurts model ability for solving problems**. As illustrated in Table 1, incorporating a greater proportion of diversifying prompts with diversity-based rewards does not necessarily lead to improved RL training outcomes. In both natural and formal language reasoning tasks, models trained with a 4:1 ratio of original to diversifying RL prompts outperform those trained with a 2:1 ratio. One possible explanation for this phenomenon is that the test set prompts follow the same format as the original RL prompts, which differ structurally from the diversifying RL prompts. Consequently, excessive fine-tuning on the diversifying RL prompts may reduce the model's performance on test prompts that resemble the original RL prompt format.

## 6 CONCLUSION

We introduced **Diverse-Exploration Reinforcement Learning (DeRL)**, a lightweight, model-agnostic modification to standard reinforcement learning pipeline that augments answer-correctness with a binary diversity reward. DeRL directly tackles the exploration–exploitation imbalance that plagues existing post-training pipelines for mathematical reasoning: by penalizing repetitions of an earlier chain-of-thought (CoT) in the natural language reasoning task or the overuse of automation tactics in the formal lanugage reasoning task, it steers the policy toward genuinely novel solution paths while still respecting ground-truth correctness. Empirically, DeRL delivers consistent gains across both natural-language and formal-language settings. With 7B-parameter Qwen, DeepSeek, and Goedel models, a 4:1 mix of standard to diversifying prompts yields more than **10 % relative improvement** in Pass@1 on the MATH and Leanabell benchmarks and lifts performance at higher shot counts *Pass@N* as well. During the training, we observe that the diversity of the model output increases due to the diversity metric used in DeRL. Our experiment results confirm that DeRL raises model reasoning capabilities through encouraging more diverse exploration during RL training.

## 7 LIMITATIONS & FUTURE WORK.

DeRL is designed to be orthogonal to the specific choice of reinforcement learning algorithm. The method, however, relies on an external LLM-based judge for reward evaluation, which inevitably introduces non-trivial latency and computational overhead. In an ideal setting, a comprehensive evaluation would consider both PPO and GRPO across models of varying scales to more fully characterize the benefits of DeRL. Nevertheless, the substantial cost associated with repeated external judge queries, combined with the practical constraints of training only up to 7B-parameter models, leads us to restrict our current experiments to use PPO only for training models with 7B scale.

Scaling DeRL to larger model checkpoints and integrating more cost-efficient external judges represent promising directions for future work. Beyond the domain of mathematical reasoning, we also plan to extend DeRL to broader application areas such as multi-step tool use, program synthesis, and open-domain dialogue—settings in which exploration bottlenecks continue to pose significant challenges for existing approaches. Another question to be studied is to enforce semantic dissimilarity not only from the initial chain-of-thought (CoT), as explored in this work, but also from successive generations. Specifically, one can instruct the model to produce a solution that is dissimilar to all the previous $n$ tries during RL training. While diminishing returns are expected as $n$ increases, it remains an open question at which point additional generations cease to provide meaningful diversity gains.

## 8 APPENDIX: CASE STUDIES

Here we show three cases for DeRL. For each case, there are three boxes. The first box includes the original RL prompts and the LLM solution. The second box includes the diversifying RL prompts and the LLM solution with respect to the diversifying prompt. The third box includes the proof plan, LLM judge prompt and the LLM judge output.

### 8.1 CASE 1

---

**LLM solution for the original RL prompt**

**Original RL prompt:**
Count how many whole numbers divide 50 without a remainder. Now let's solve the problem step by step.

**LLM Solution:**
A positive integer divisor of 50 is a positive integer that divides 50 evenly, i.e., without leaving a remainder. To find the divisors of 50, we can list them out in pairs that multiply to 50:

- 1 and 50
- 2 and 25
- 5 and 10

We don't consider 10 and 5 as separate pairs because 10 is already in the list.
So, there are $\boxed{6}$ positive divisors of 50: 1, 2, 5, 10, 25, 50.

---

**LLM solution for the diversifying RL prompt**

**Diversifying RL prompt:**
Problem statement: Count how many whole numbers divide 50 without a remainder. I want you to solve the problem in a way that is very dissimilar to this proof plan. If you solve the problem in a way similar to the proof plan, that is cheating.
Proof plan starts: To find the number of divisors, identify all pairs of numbers that multiply to give 50. This involves writing down the factors in pairs, ensuring no number is counted twice. The final count is these pairs combined. Proof plan ends.
Now let's solve the problem step by step.

**Student solution:**
First, we need to find the prime factorization of 50.
50 can be written as $2 \cdot 5^2$.
Now, we need to count the number of divisors.
For a number with prime factorization $p_1^{e_1} \cdot p_2^{e_2} \cdots p_n^{e_n}$, the total number of divisors is $(e_1 + 1)(e_2 + 1) \cdots (e_n + 1)$.
In this case, the total number of divisors is $(1 + 1)(2 + 1) = 2 \cdot 3 = 6$.
So, the total number of whole numbers that divide 50 without a remainder is 6.
The answer is $\boxed{6}$.

---

**LLM judge for the diversity metric**

**Proof plan in the diversifying prompts:**
To find the number of divisors, identify all pairs of numbers that multiply to give 50. This involves writing down the factors in pairs, ensuring no number is counted twice. The final count is these pairs combined.

**Judge Prompt:**
You are a careful math teacher. I will give you the problem statement, the solution of the student, and a proof plan.
The student has been taught with the proof plan, and he is instructed to find a method to solve the problem that is very different from the proof plan.
I want you to carefully check if the solution of the student is very similar to the proof plan.

**Problem statement:** {prompt}
**Proof plan:** {proof_plan}
**Student solution:** {solution_str}

Output ***True*** if the solution is similar to the proof plan (so the student cheated). In all other cases, output ***False***.

**Judge output:**
After analyzing the student's solution and the proof plan:

Output: [ False ]

The student's solution is fundamentally different from the proof plan. The proof plan suggests a direct approach of identifying and listing all pairs of numbers that multiply to give 50. This would involve manually finding factors like $1 \times 50$, $2 \times 25$, $5 \times 10$.
The student instead used a more algebraic method based on prime factorization. They first found the prime factorization of 50 as $2 \cdot 5^2$, then applied the formula $(e_1+1)(e_2+1)\cdots(e_n+1)$ to calculate the number of divisors. This method uses exponents and a general formula rather than direct enumeration of the factors.

## 8.2 CASE 2

**LLM solution for the original RL prompt**

**Original RL prompt:**
Given vectors $\mathbf{a}$, $\mathbf{b}$, and $\mathbf{c}$ with magnitudes 1, 2, and 3 respectively, if the sum of their cross products
$$\mathbf{a} \times \mathbf{b} + \mathbf{b} \times \mathbf{c} + \mathbf{c} \times \mathbf{a} = \mathbf{0},$$
find the value of the scalar triple product $\mathbf{a} \cdot (\mathbf{b} \times \mathbf{c})$. Now let's solve the problem step by step.

**LLM Solution:**
The scalar triple product $\mathbf{a} \cdot (\mathbf{b} \times \mathbf{c})$ equals the volume of the parallelepiped determined by $\mathbf{a}$, $\mathbf{b}$, and $\mathbf{c}$.
Since
$$\mathbf{a} \times \mathbf{b} + \mathbf{b} \times \mathbf{c} + \mathbf{c} \times \mathbf{a} = \mathbf{0},$$
this is a known identity that holds when the three vectors lie in the same plane, i.e., they are coplanar.
Coplanar vectors form a parallelepiped of zero volume. Therefore,
$$\mathbf{a} \cdot (\mathbf{b} \times \mathbf{c}) = \boxed{0}.$$

## LLM solution for the diversifying RL prompt

**Diversifying RL prompt:**
Given vectors $\mathbf{a}$, $\mathbf{b}$, and $\mathbf{c}$ with magnitudes 1, 2, and 3 respectively, if the sum of their cross products
$$\mathbf{a} \times \mathbf{b} + \mathbf{b} \times \mathbf{c} + \mathbf{c} \times \mathbf{a} = \mathbf{0},$$
find the value of the scalar triple product $\mathbf{a} \cdot (\mathbf{b} \times \mathbf{c})$.

I want you to solve the problem in a way that is very dissimilar to this proof plan. If you solve the problem in a way similar to the proof plan, that is cheating.

Proof plan starts: The scalar triple product represents the volume of a parallelepiped formed by the three vectors. The given condition implies that the three vectors are coplanar. Then we are essentially finished.
Proof plan ends

Now let's solve the problem step by step.
**Student solution:**
We start from the identity:

$$\mathbf{a} \times \mathbf{b} + \mathbf{b} \times \mathbf{c} + \mathbf{c} \times \mathbf{a} = \mathbf{0}.$$

Rewriting,
$$\mathbf{a} \times \mathbf{b} = -\mathbf{b} \times \mathbf{c} - \mathbf{c} \times \mathbf{a}.$$

Take the dot product of both sides with $\mathbf{a}$:

$$\mathbf{a} \cdot (\mathbf{a} \times \mathbf{b}) = -\mathbf{a} \cdot (\mathbf{b} \times \mathbf{c}) - \mathbf{a} \cdot (\mathbf{c} \times \mathbf{a}).$$

Now, note that $\mathbf{a} \cdot (\mathbf{a} \times \mathbf{b}) = 0$ since any vector dotted with its cross product with another vector is zero.
Also, $\mathbf{a} \cdot (\mathbf{c} \times \mathbf{a}) = 0$ for the same reason.
This simplifies the equation to:
$$0 = -\mathbf{a} \cdot (\mathbf{b} \times \mathbf{c}),$$

so we conclude:
$$\mathbf{a} \cdot (\mathbf{b} \times \mathbf{c}) = \boxed{0}.$$

> **LLM judge for the diversity metric**
>
> **Proof plan in the diversifying prompts:**
> The scalar triple product represents the volume of a parallelepiped formed by the three vectors. The given condition implies that the three vectors are coplanar. Then we are essentially finished.
>
> **Judge Prompt:**
> You are a careful math teacher. I will give you the problem statement, the solution of the student, and a proof plan.
> The student has been taught with the proof plan, and he is instructed to find a method to solve the problem that is very different from the proof plan.
> I want you to carefully check if the solution of the student is very similar to the proof plan.
>
> **Problem statement:** {prompt}
> **Proof plan:** {proof_plan}
> **Student solution:** {solution_str}
>
> Output ***True*** if the solution is similar to the proof plan (so the student cheated). In all other cases, output ***False***.
>
> **Judge output:**
> After carefully comparing the student's solution to the proof the student was taught with, I can determine that:
> ***False***
> The student's solution uses a significantly different approach from the taught proof. The taught proof relies on the geometric interpretation of the scalar triple product, showing that the vectors are coplanar and therefore the volume of the parallelepiped is zero.
> In contrast, the student's solution uses algebraic manipulation of vector identities. They: 1. Rearranged the given equation 2. Took the dot product with vector a 3. Applied the vector triple product identity 4. Analyzed orthogonality relationships

## 8.3  CASE 3

Warning: the ground truth provided in `nvidia/OpenMathInstruct-2` for this problem is wrong (the answer is not 1).

**LLM solution for the original RL prompt**

**Original RL prompt:**
Calculate the value of $\frac{(a+b)^2(a-b)^2(a^2+b^2)^2}{(a^2-b^2)^4}$ where $a$ and $b$ are non-zero real numbers. Now let's solve the problem step by step.

**LLM Solution:**
We can start by simplifying the expression step by step:

$$
\begin{aligned}
\frac{(a+b)^2(a-b)^2(a^2+b^2)^2}{(a^2-b^2)^4} &= \frac{(a^2+2ab+b^2)(a^2-2ab+b^2)(a^4+2a^2b^2+b^4)}{(a^4-2a^2b^2+b^4)^2} \\
&= \frac{[(a^2+b^2)+2ab][(a^2+b^2)-2ab](a^2+b^2)^2}{(a^2+b^2)^2(a^2-b^2)^2} \\
&= \frac{(a^2+b^2+2ab)(a^2+b^2-2ab)}{(a^2-b^2)^2} \\
&= \frac{(a^2+2ab+b^2)(a^2-2ab+b^2)}{(a^2-b^2)^2} \\
&= \frac{(a+b)^2(a-b)^2}{(a+b)^2(a-b)^2} \\
&= \boxed{1}.
\end{aligned}
$$

So the value of the expression is $\boxed{1}$.

**LLM solution for the diversifying RL prompt**

**Diversifying RL prompt:**

Problem statement: Calculate the value of $\frac{(a+b)^2(a-b)^2(a^2+b^2)^2}{(a^2-b^2)^4}$ where $a$ and $b$ are non-zero real numbers.

I want you to solve the problem in a way that is very dissimilar to this proof plan. If you solve the problem in a way similar to the proof plan, that is cheating.

Proof plan starts:

The solution involves algebraic manipulation of the expression. First, we expand some terms in the numerator. Then we recognize that $(a^2 - b^2)^2$ appears in both numerator and denominator, allowing for simplification. We also identify that $(a^2 + b^2)^2$ in the numerator helps simplify the expression.

Proof plan ends.

Now let's solve the problem step by step

**Student solution:**

The denominator is a difference of squares:

$$(a^2 - b^2)^4 = [(a+b)(a-b)]^4 = (a+b)^4(a-b)^4.$$

Now notice that $(a+b)^4$ and $(a-b)^4$ are also parts of the numerator. In fact, the numerator to this point is

$$(a+b)^2(a-b)^2(a^2+b^2)^2 = (a+b)^2(a-b)^2(a^2+b^2)^2,$$

and multiplying these out gives:

$$(a+b)^4(a-b)^4(a^2+b^2)^2 = \text{denominator} \cdot (a^2+b^2)^2.$$

So we have:

$$\frac{(a+b)^2(a-b)^2(a^2+b^2)^2}{(a+b)^4(a-b)^4} = \frac{(a+b)^2(a-b)^2}{(a+b)^2(a-b)^2} \cdot \frac{(a^2+b^2)^2}{(a+b)^2(a-b)^2}.$$

The first fraction on the right simplifies to 1, and we notice that the denominator of the second fraction is the same as what we just canceled (since it's multiplied by $(a^2+b^2)^2$ in the numerator). So their quotient is 1.

Thus, the final answer is:

$$1 \cdot 1 = \boxed{1}.$$

810
811
812
813
814
815
816
817
818
819
820
821
822
823
824
825
826
827
828
829
830
831
832
833
834
835
836
837
838
839
840
841
842
843
844
845
846
847
848
849
850
851
852
853
854
855
856
857
858
859
860
861
862
863

---

**LLM judge for the diversity metric**

**Proof plan in the diversifying prompts:**
The solution involves algebraic manipulation of the expression. First, we expand some terms in the numerator. Then we recognize that $(a^2 - b^2)^2$ appears in both numerator and denominator, allowing for simplification. We also identify that $(a^2 + b^2)^2$ in the numerator helps simplify the expression.

**Judge Prompt:**
You are a careful math teacher. I will give you the problem statement, the solution of the student, and a proof plan.
The student has been taught with the proof plan, and he is instructed to find a method to solve the problem that is very different from the proof plan.
I want you to carefully check if the solution of the student is very similar to the proof plan.

**Problem statement:** {prompt}
**Proof plan:** {proof_plan}
**Student solution:** {solution_str}

Output ***True*** if the solution is similar to the proof plan (so the student cheated). In all other cases, output ***False***.

**Judge output:**
After analyzing the student's solution and the proof plan: Output: | False |

After carefully reviewing both the proof plan and the student's solution, I can conclude:
The student's solution uses a fundamentally different approach from the proof they were taught with. The taught proof expands the algebraic expressions and simplifies them step by step, whereas the student's solution takes a different path by:

1. Recognizing factorization patterns in the denominator: $(a^2 - b^2)^4 = [(a+b)(a-b)]^4$

2. Comparing these factorization patterns with what appears in the numerator

3. Dividing the expression into two separate fractions to show the value equals 1

Though both solutions arrive at the same answer (1), the student's reasoning process and algebraic manipulations are distinctly different from the taught proof.

---

## 9    THE USE OF LLM

ChatGPT 5 has been used in polishing the writing for the paper and finding related work.

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
