# OpenReview forum: "DeRL: Diverse‑Exploration Reinforcement Learning for Large Language Models Improves Mathematical Reasoning"
_ICLR.cc/2026/Conference — ICLR 2026 Conference Withdrawn Submission_

### Official Review · Reviewer_F2oC · 2025-10-31

**Soundness:** 3
**Presentation:** 3
**Contribution:** 2
**Rating:** 4
**Confidence:** 4

**Summary:**

This paper introduces Diverse-Exploration Reinforcement Learning (DeRL), a modification to standard RL pipelines for training LLMs on mathematical reasoning tasks. The authors identify a key limitation in current RL approaches: models tend to over-exploit a few high-probability chain-of-thought (CoT) sequences, neglecting the rich space of alternative solution strategies. DeRL addresses as follows.For natural language math problems: explicitly instructing the model to solve problems differently from previously generated CoTs, with an auxiliary LLM judge (Claude-3.7) verifying dissimilarity. For formal theorem proving in Lean: penalizing the use of common automation tactics (linarith, omega, simp, etc.).

The diversity metric is combined with correctness to create a reward signal that encourages exploration while maintaining accuracy. Experiments on 7B models (Qwen, DeepSeek, Goedel) show consistent improvements over PPO baselines: >10% relative gain on Pass@1 for MATH and Leanabell benchmarks, with better Pass@N performance and progressively increasing solution diversity during training.

**Strengths:**

Originality: While diversity-aware RL exists, the specific application to CoT reasoning with LLM judges is novel. The identification of automation tactic overuse in Lean provers is an original contribution.

Quality: Consistent improvements shown across 3 different models and 2 distinct task types. Both Pass@1 and Pass@N metrics, plus diversity tracking during training are shown. Appropriate baselines used, that include random reward experiments to validate that improvements are not spurious. The "Non-LLM-Rollout" experiment demonstrates the importance of using the learner's own CoTs.

Clarity: The over-exploitation issue is clearly articulated with concrete examples.

Significance: The exploration-exploitation imbalance is an important limitation in current RLHF pipelines. >10% improvements are meaningful for downstream applications.

**Weaknesses:**

1. Major Issues

1.1 Limited scalability analysis: Only 7B models tested; unclear if benefits persist at larger scales. External LLM judge (Claude-3.7) creates significant API costs and latency. Authors acknowledge this but do not provide cost analysis or alternatives.

1.2 Shallow diversity analysis: LLM judge evaluates "semantic similarity" but this is a black box. No analysis of whether diverse solutions represent different mathematical insights vs. superficial variations. Case studies show solutions that arrive at the same answer via similar logic, are they truly diverse?

1.3 Incomplete comparisons: No comparison to other diversity-promoting RL methods (entropy regularization, curiosity-driven learning, etc.). Missing comparison to GRPO with proper hyperparameter tuning. The claim that DeRL is "orthogonal to RL algorithm" is asserted but not empirically validated beyond PPO.

1.4 Experimental limitations: Only 1 epoch of training, what happens with continued training? The 4:1 ratio appears optimal but limited exploration of this hyperparameter space. Test set may have distribution shift from training (different prompt formats).


2. Minor Issues

2.1 Methodological concerns: The "proof plan" extraction using DeepSeek-R1 adds confounds, what if the summarization is poor?
Automation tactic blacklist seems ad-hoc; no principled way to select which tactics to forbid. Case 3 in Appendix notes the ground truth is wrong, how common is this in the dataset?

2.2 Presentation: The distinction between "original RL prompts" and "diversifying RL prompts" could be clearer upfront.

2.3 Statistical rigor: No error bars or significance tests reported. Single run per configuration (?). "Best checkpoint" selection could introduce selection bias.

2.4 Theoretical gap: Limited theoretical justification for why diversity should improve Pass@1 (not just Pass@N). No analysis of the diversity-accuracy tradeoff.

**Questions:**

Main questions

1. On LLM Judge reliability:

1.1 What is the inter-rater reliability between Claude-3.7 and other judges (e.g., GPT-4, human annotators)?

1.2 How sensitive are results to the specific judge prompt? Were alternative prompts tested?

1.3 Can you quantify cases where the judge incorrectly labels dissimilar solutions as similar (false positives) or vice versa?

2. On true mathematical diversity:

2.1 Can you provide a more rigorous taxonomy of what constitutes "diverse" mathematical approaches (e.g., algebraic vs. geometric, constructive vs. proof-by-contradiction)?

2.2 How do you distinguish between genuinely different solution strategies vs. superficial lexical variations?

2.3 In the case studies, both solutions use similar high-level strategies, are they mathematically distinct?

3. On scalability:

3.1 What is the cost-benefit analysis of using Claude-3.7 as a judge? (API costs vs. performance gains)

3.2 Have you experimented with a smaller, fine-tuned judge model to reduce costs?

3.3 Have you experimented if benefits persist with larger base models (70B+)?

4. On generalization:

4.1 Does DeRL help on out-of-distribution problems (e.g., MATH test set problems significantly harder than training)?

4.2 How does performance change on the test set throughout training (not just at best checkpoint)?


Clarification questions

5.1 Experimental details: How many training runs were performed? Are results from single runs or averaged?
What is the compute budget comparison between DeRL and baseline PPO (given the LLM judge calls)?
Why only 1 epoch? What happens with continued training?

5.2 On the 4:1 ratio: You show 2:1 and 4:1, but what about 8:1 or other ratios? Is there a principled way to select this?
Does the optimal ratio depend on problem difficulty or model capacity?

5.3 On automation tactics: How were the 9 blacklisted tactics selected? What about other powerful tactics like tauto, decide, norm_num?
Do models learn to avoid these tactics even on original RL prompts after DeRL training?

5.4 On random rewards: The random reward results are confusing, why does it help Qwen but hurt DeepSeek?
This seems to undermine the claim that diversity rewards are specifically valuable.

---

### Official Review · Reviewer_rAjY · 2025-11-02

**Soundness:** 1
**Presentation:** 2
**Contribution:** 1
**Rating:** 2
**Confidence:** 4

**Summary:**

The paper proposes DeRL, a diversity‑encouraging RL fine‑tuning method for reasoning LLMs. In addition to a standard correctness reward, DeRL introduces diversifying RL prompts plus a diversity reward: for natural‑language math, an external LLM judge labels whether a new chain of thought is dissimilar to a prior answer; for Lean, the diversity reward withholds credit if certain tactics (e.g., linarith, simp, omega) appear. DeRL is evaluated on MATH (with Qwen‑7B, DeepSeek‑7B) and Leanabell (with Goedel‑Prover).

**Strengths:**

The method is straightforward and easy to understand.

**Weaknesses:**

- The paper does not situate DeRL against recent exploration‑oriented RL for reasoning (e.g., Cheng et al., 2025), nor does it compare to alternative exploration rewads (e.g., entropy‑regularized reward or Pass@K‑aware training (Walder & Karkhanis, 2025)).

- Across all three settings, the largest reported Pass@1 gain over standard PPO is 0.013 (0.417 → 0.430 on DeepSeek‑MATH). Qwen‑MATH improves 0.009 (0.614 → 0.623), and Leanabell improves 0.011 (0.767 → 0.778) (Table 1). The paper does not specify how many random seeds were used, which further diminishes the significance of the results. The Pass@N curves for MATH (Fig. 2) likewise show small gaps that do not convincingly link “more diversity” to better task performance.

- The primary diversity evidence is measured on diversifying prompts with an external judge (Fig. 3a–b) or via a tactic blacklist in Lean (Fig. 3c). It remains unclear whether the base model could reach similar diversity without the diversity reward simply by using these diversifying prompts, or whether diversity improves under standard prompts (the ones used for evaluation).

Cheng, Daixuan, et al. "Reasoning with exploration: An entropy perspective." arXiv preprint arXiv:2506.14758 (2025).

Walder, Christian, and Deep Karkhanis. "Pass@ K Policy Optimization: Solving Harder Reinforcement Learning Problems." arXiv preprint arXiv:2505.15201 (2025).

**Questions:**

See Weaknesses.

---

### Official Review · Reviewer_kyGY · 2025-11-03

**Soundness:** 2
**Presentation:** 2
**Contribution:** 2
**Rating:** 2
**Confidence:** 4

**Summary:**

The paper introduces a pipeline DeRL for diverse-exploitation RL, which uses a history solve path as a prompt to ask the model to generate a different proof path explicitly. The method improves upon a standard PPO baseline in both the natural language math problem-solving task and the formal task.

**Strengths:**

- The paper investigates an important aspect of RL learning. where the diversity of the model's response will naturally decrease during the RL training.

**Weaknesses:**

- The method for natural language and formal environment is totally detached. I don't see the point of having a formal part of this paper, as the method only uses a list of prohibited tactics. The evaluation for the formal part is also quite weak, using a learnable test set. It's better to use miniF2F or Putnam-bench for a more robust evaluation.

- The proposed method is somewhat trivial and unscalable. The improvement it brings is also very minor. I don't think this method can cause any broader impact.

**Questions:**

- If the cost is the major reason for using PPO, you can also use Qwen instead with GRPO. The Judging task doesn't seem to be very complicated and is totally capable of being determined by Qwen models.
- Instead of comparing a standard PPO baseline, compare with other diversified RL methods in the fields like  Pass@k Training.
- The formal part is not necessary, and I don't see how it can provide any more insight.

---

### Note · Authors · 2025-12-03

I have read and agree with the venue's withdrawal policy on behalf of myself and my co-authors.